# Structural evidence for extracellular silica formation by diatoms

Boaz Mayzel [1,4], Lior Aram [1,4], Neta Varsano[2], Sharon G. Wolf [3] & Assaf Gal [1✉]

The silica cell wall of diatoms, a widespread group of unicellular microalgae, is an exquisite example for the ability of organisms to finely sculpt minerals under strict biological control. The prevailing paradigm for diatom silicification is that this is invariably an intracellular process, occurring inside specialized silica deposition vesicles that are responsible for silica precipitation and morphogenesis. Here, we study the formation of long silicified extensions that characterize many diatom species. We use cryo-electron tomography to image silica formation in situ, in 3D, and at a nanometer-scale resolution. Remarkably, our data suggest that, contradictory to the ruling paradigm, these intricate structures form outside the cytoplasm. In addition, the formation of these silica extensions is halted at low silicon concentrations that still support the formation of other cell wall elements, further alluding to a different silicification mechanism. The identification of this unconventional strategy expands the suite of mechanisms that diatoms use for silicification.

[1] Department of Plant and Environmental Sciences, Weizmann Institute of Science, Rehovot, Israel. [2] Department of Structural Biology, Weizmann Institute of Science, Rehovot, Israel. [3] Department of Chemical Research Support, Weizmann Institute of Science, Rehovot, Israel. [4]These authors contributed equally: Boaz Mayzel, Lior Aram. ✉email: assaf.gal@weizmann.ac.il

Diatoms are a highly abundant group of unicellular photosynthetic organisms[1]. The hallmark of diatoms is their ability to synthesize a delicately sculptured cell wall made of silica ($SiO_2*H_2O$)[2,3]. The cell wall comprises an array of several distinct elements, cemented together to form a continuous coverage. The formation process of the cell wall is under strict biological control, giving rise to species-specific nanoscale architectural motifs[4–7].

How do diatoms control the precipitation process of the silica elements? This question has intrigued scientists for over a century, in the context of both a model system for biomineralization and a source of inspiration for biomimetic production of nanomaterials for biotechnology[8,9]. Although it is known that other organisms can form silica extracellularly[10,11], the current view on diatom silicification is that it is invariably an intracellular process[2,3]. According to this paradigm, specialized micron-sized compartments within the diatom cell, termed silica deposition vesicles (SDVs), are responsible for the entire mineralization and morphogenesis processes of the silica elements. Only upon completion, a silica element is exocytosed and appended to the existing cell wall. The concept of intracellular silicification serves as the basis for explaining fundamental traits of diatoms, such as the observed cell-size reduction following cytokinesis and the outstanding morphological control over the silica precipitation process[12,13].

Interestingly, several diatom species have elaborate silica appendages that can be much longer than the cell body. Such extensions challenge the concept of intracellular silicification, with some resemblance to the extremely long spicules formed by sponges in a process that is proposed to start intracellularly but is predominantly extracellular[14]. This problem has been scarcely addressed in previous works, mostly due to the difficulty in studying the formation process of such elements. A striking example for such long elements are whisker-like extensions, called setae, which characterize the genus *Chaetoceros*[15,16]. Each seta can reach a length of tens of micrometers, far exceeding the length of the cell body (Fig. 1).

Here, we study the formation mechanism of setae in the species *Chaetoceros tenuissimus* using cryo-electron tomography (cryoET). The high-resolution native-state information acquired using cryoET clearly resolve cellular structures such as lipid bilayer membranes. Surprisingly, the seta silica is always present outside the cell membrane, suggesting a formation process that is not mediated by an SDV. In addition, this process is more susceptible to low-Si medium than other silica elements that grow inside SDVs. Such a highly controlled extracellular silicification process in diatoms raises many interesting questions regarding the fundamentals of biological silicification.

## Results

*C. tenuissimus* is a relatively small barrel-shaped diatom species, with a cell size of 5–7 μm (Fig. 1a, c). Two long setae grow at an angle of ~45° to the apical axis from opposite locations on the rim of each valve (the circular 'lid' of the cell wall)[17]. The setae are up to 20 μm in length and have a narrow diameter of ~300 nm (Fig. 1b). Each seta is enveloped by a silica structure constructed of helical longitudinal rods and short transversal elements, which connect the longer rods (Fig. 1d). Regularly, short silica spines protrude from the seta, but the entire length of the helical structure is continuous without any observable sutures. Such sutures characterize the instances where two distinct silica elements are cemented at the cell wall[18–20].

In order to facilitate structural investigations into the biological process of seta silicification, we adjusted established procedures, used for other diatom species, to synchronize the cell cycle in *C. tenuissimus* cultures[21]. We exposed exponentially growing cultures to a short period of silicon-starvation, which resulted in cell cycle arrest. Upon silicon replenishment, the cells resumed their growth and silicification in a partially synchronized fashion. This was corroborated by staining the cells with the fluorescent dye PDMPO, which is incorporated only into newly formed silica[22]. We estimate that about 50% of the cells were actively forming their new valves and setae six hours after silicon addition (Fig. 2a–c). It was previously shown for different *Chaetoceros* species that the setae elongate from their tip[15,23]. In accordance with such tip growth process, we could sometime observe the furthermost part of the seta stained with PDMPO when the dye was added for a couple of minutes to a synchronized culture at the stage of seta formation (Fig. 2d, e). Overall, we observed that the formation process of setae is very rapid and lasts for about 20 min.

We developed a workflow to study the process of seta formation at molecular resolution by preparing actively silicifying cells for cryoET. The cell cycle of a *C. tenuissimus* culture was synchronized, and at the time point when a maximal fraction of the cells were at the stage of seta formation, cell samples were vitrified by plunging them into liquid ethane. For data collection, we chose cells that had two long setae (mature and belonging to the old valve), and two short setae (forming at the new valve), and collected data from both types of setae of such cells (Fig. 3a). This approach proved more practical than correlative PDMPO fluorescence and electron microscopy for locating forming setae. The 2D cryo-electron microscopy images of frozen setae show the overall structure of the silica element (Fig. 3b). In order to study the structural organization within the seta, we collected tilt-series images and reconstructed the 3D volume information. Figure 3c–i and Supplementary Movie 1 show both 2D slices within the reconstructed volume and 3D volume renderings for an actively silicifying seta tip. The helical arrangement of the silica cell wall is clearly shown (Fig. 3g, h, Supplementary Movie 1). Interior to the cell wall, the cell membrane is apparent, containing a single long microtubule element and sometimes a few membranous vesicles

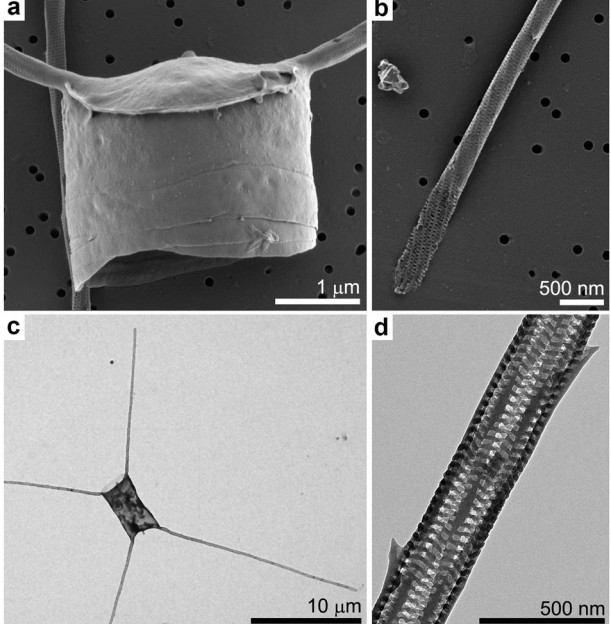

**Fig. 1 The silica cell wall morphology of *C. tenuissimus*. a**, **b** Scanning electron microscopy (SEM) images of cells dried on a support membrane. **c**, **d** Transmission electron microscopy (TEM) images of cell walls washed in water and air dried onto a grid.

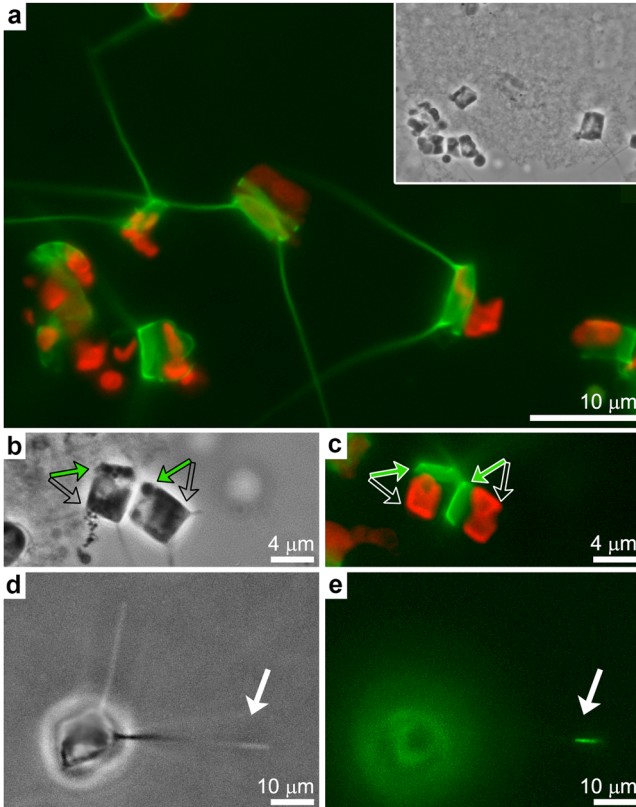

**Fig. 2 Synchronization of seta formation. a** A synchronized *C. tenuissimus* culture, where PDMPO fluorescence (green) is observed only in one half of the cell wall. Chlorophyll autofluorescence is in red. Inset shows a bright-field image of the cells. **b, c** Close-up view of two synchronized cells. Arrows show the newly formed and old valves (green and transparent arrows, respectively). **d, e** A single cell at the stage of seta formation immediately after the addition of PDMPO. Only the seta tip is labeled with PDMPO (arrows), indicating that this is the forming end of the seta. The rest of the cell is out of focus.

(Fig. 3c, e, Supplementary Movie 1). Interestingly, external to the silica structure we observed another thin layer that is continuous and covers the entire seta (Fig. 3f). This layer is unilamellar and thinner than a lipid bilayer (Fig. 3i), and may be a polysaccharide envelope that is associated with many diatom species[20,24].

This simple cellular arrangement, namely a single microtubule within a cytoplasmic extension with extracellular silica and an organic coat, is shared by almost all the short setae we investigated (Table 1, Supplementary Fig. 1). This is important, as previous structural investigations using chemically fixed samples concluded that the end of a growing seta is open and not covered by silica[16]. In our data, a small number of short setae showed such an open morphology, which was an artifact of mechanical damage during the sample preparation process. In some of these damaged seta tips, we could identify distinct fibrous elements, external to the cell membrane and aligned with the forming silica rods (Fig. 4a). We speculate that such fibers might belong to an organic scaffold that is guiding silica precipitation.

When long setae were imaged and analyzed, a very different cellular arrangement was observed (Table 1, Supplementary Fig. 1). These fully formed setae belong to the older half of the cell wall and serve as an internal control demonstrating the cellular arrangement after silica formation. The cytoplasm was usually fragmented into vesicles and detached sheets (Fig. 4b), the microtubule was distorted, and in several instances only some

debris was seen within the seta interior (Fig. 4c). This implies that once seta growth is completed, the active cytoplasm retracts into the main cell body.

The absence of an SDV in the cryoET observations, at spatial and temporal resolutions that are sufficient to identify such a compartment if present, provides a structural evidence that points to a silification process that does not involve an SDV. Since the fact that we did not observe an SDV is not logically sufficient to prove its absence, we sought more evidence regarding the process of seta formation. We investigated the effects of low Si concentrations in the growth medium, as well as silification inhibitors, on the silification process in these cells. It is expected that silica elements that form inside the intracellular environment of an SDV, e.g., the valves, will be more resilient to such environmental challenges than the setae that form in an extracellular environment, although controlled and confined.

We grew *C. tenuissimus* cultures at controlled and constant chemical conditions in seawater media that varied in their Si concentrations. As the concentration of Si was reduced, the growth rate of the cells slowed down accordingly, from 2.6 cell divisions per week at 330 μM Si of the replete conditions, to 2.1 cell divisions per week at 10 μM Si, to 1.6 cell division per week at 5 μM Si and 1.7 cell divisions per week at 2.5 μM Si. The morphology of the cell wall at the different Si concentrations was investigated by both optical and electron microscopy. At Si concentrations of 10 μM and higher, only very minor modifications were observed, but at the very low Si concentrations of 5 μM and 2.5 μM, a growing fraction of the cells had shorter setae or lacked them altogether (Fig. 5a–c). As noted above, the cells reproduced even at these low Si concentrations, and continued to form their valves and other SDV-related cell wall elements. We used the contrast of the cell wall in high-angle annular dark-field scanning transmission electron microscopy (HAADF-STEM) imaging mode as a proxy for the Si content of the cell wall[25,26]. These images show a minor reduction in the contrast of the dried cells as the Si concentration of the medium is reduced (Fig. 5e–g). In order to relate the HAADF-STEM contrast to Si content in the valve, the cell walls were extracted by a mild oxidizing treatment that removes adherent organic material. A quantitative HAADF-STEM imaging of such cell walls indeed shows a minor reduction of the silica content of the valves as Si concentrations of the medium were lowered (Supplementary Fig. 2).

A similar effect was observed when germanium (Ge), a Si analog that inhibits diatom silification, was added to the growth media. At high concentrations of Ge cell wall formation arrests, probably due to inhibition of Si uptake by the cell[6]. Nevertheless, we found that at intermediate Ge concentration of 10 μM, the cells managed to complete at least one cell cycle. In order to identify the specific valves that were produced in the presence of Ge, PDMPO was added to the culture concurrently with Ge addition. PDMPO labeled valves were located on the TEM grid using a correlative optical-electron microscopy approach (Fig. 5d, h insets). These valves lacked setae altogether and showed malformed silica texture (Fig. 5d, h). Together, these two experiments demonstrate differential influence of silification-related environmental conditions on the formation process of setae vs. other cell wall components, in agreement with the cryoET results.

We noticed that the effect of low Si concentration took several generations to become evident in the growth experiments, and was very mild after the first cell division in the new conditions. Cells that were exposed to low Si for only a short time produced setae that were very similar to the ones produced at Si-replete conditions, both in their length and in their Si content (Supplementary Fig. 3). In addition, the Si content of setae that formed at long-term cultivation in lower Si concentrations was similar to that of setae formed at Si-replete conditions, as evaluated using

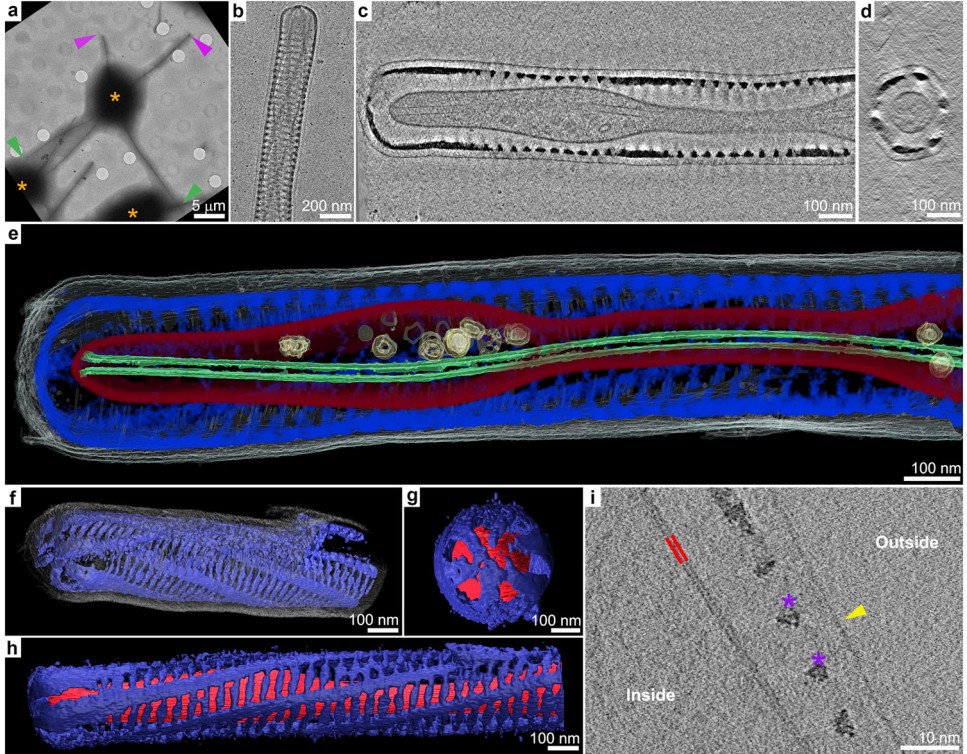

**Fig. 3 3D structure of actively growing seta. a** A low-magnification cryoTEM image of vitrified *C. tenuissimus* cells. The thick cell bodies are indicted with (*). The middle cell has two fully formed setae (green arrowheads) and two shorter, presumably forming, setae (purple arrowheads). **b** A high-magnification cryoTEM projection image of the tip of a forming seta. **c, d** 2D orthogonal views through the 3D reconstructed tomogram of a forming seta. **e** Volume rendering of the different components of a growing seta: The microtubule is in green, intracellular vesicles in yellow, cell membrane in red, silica cell wall in blue, and external layer in gray. **f** 3D view of the silica cell wall and external layer only. **g, h** 3D views of the silica cell wall and cell membrane only. **i** Zoom in on a slice through a 3D data set, demonstrating the structural difference between the lipid bilayer membrane of the cell (in red) and the envelope external to the cell wall (yellow arrowhead); silica is indicated by (*). See Supplementary Movie 1 for an animation of the 3D reconstruction.

**Table 1 The structural characteristics of analyzed setae.**

| Seta length | Continuous microtubule | Intact cell membrane | Fragmented or absent cell membrane | # of observed cells |
|---|---|---|---|---|
| Short (<10 um) | + | + | − | 9 |
| | − | + | − | 2 |
| | − | − | + | 2 |
| Long (>10 um) | + | + | − | 2 |
| | − | + | − | 2 |
| | − | − | + | 11 |

the contrast at HAADF-STEM imaging (Supplementary Fig. 3). These observations suggest that direct uptake of Si from seawater into the confined volume at the seta tip where the silica forms is not a major Si source for seta formation. Rather, the transport of Si building blocks for seta formation is likely to be via an intracellular route, where Si is concentrated by an active cellular mechanism and then transported through the cytoplasmic extension to the seta tip. Such a process can explain the delayed effect of changing the Si concentration in the medium and the regular density of the seta silica at very different Si concentrations of the medium.

## Discussion

Diatom silicification is a complex process that requires several levels of regulation, from controlling the inorganic precipitation reaction, to species-specific morphogenesis routes, to unique exocytosis events[2,3]. In this respect, studying seta formation in *C.*

*tenuissimus* gave us several important advantages. First, the thin setae of this species do not allow large cellular organelles to penetrate the seta volume, leaving a bare-bone system, where silica forms at the seta tip in the presence of only a single observable cellular element, a microtubule, and occasionally small vesicles. This is in contrast to larger *Chaetoceros* species that were studied previously, where organelles such as mitochondria within the seta, were proposed to actively participate in seta silicification[15,16]. In addition, the narrow diameter of the seta mitigated the major challenge in acquiring high-resolution cryoET data within cells, namely, sample thickness[27]. The native seta diameter of ~300 nm proved sufficiently thin to yield the needed spatial resolution. The common features between the described process and seta formation in other *Chaetoceros* species are yet to be explored.

The experimental advantages yielded the fundamental observation of a silicification process in diatoms that does not involve an SDV. It is important to stress that the resolution of cryoET in

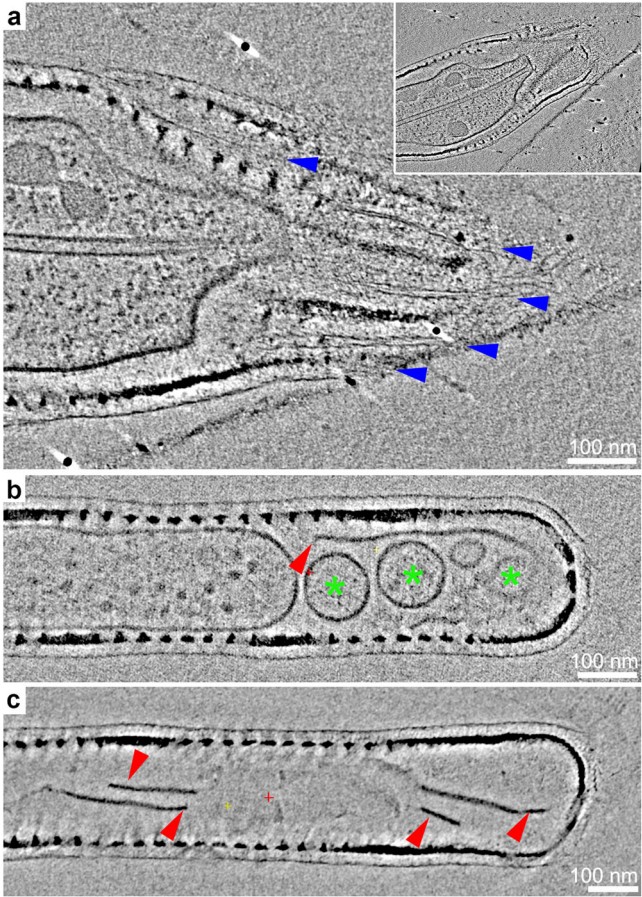

**Fig. 4 Structural properties of damaged and long setae.** Slices in the reconstructed volumes of: **a** A damaged short seta (inset shows lower magnification). Blue arrowheads point to fibrous material that is aligned with the disrupted silica rods; **b**, **c** Long setae at different stages of cytoplasm degradation. Green asterisks indicate fragmented cytoplasm and red arrowheads indicate irregular sheet-like debris.

the length and continuity of the seta. We speculate that this extracellular silicification process involves the secretion of both inorganic silica building blocks and an organic macromolecular template that is synthesized by protein machinery located at the tip of the cytoplasm membrane. Such an insoluble organic scaffold has been identified in several diatom species and resembles the process of spicule formation by sponges[14,18,28–30]. Figure 6b shows a schematic illustration of a forming seta, and Fig. 6c–f show the proposed sequential steps in seta elongation, where the organic template is continuously secreted, mineralized, and then pushed by the forward polymerization of the microtubule. An interesting consequence of such a process is that the silica rods are first mineralized in a direction perpendicular to the seta length and only later are aligned with it. This will create internal stress in the material, which may mechanically stabilize it, thus explaining why these delicate long structures do not bend even under large forces (such as the centrifugation and drying needed for sample preparation), and maintain their straight orientation[31]. Yet several points are left unanswered by this model. How do the silica elements that connect the main rods gets silicified? How does the external sheath form? What are the chemical properties of the organic scaffold? Nevertheless, we believe this model can serve as a working hypothesis to stimulate further research into the mechanism of seta formation.

An interesting open question regarding this process is the source of the silicon building blocks used to form the seta[5,6,32,33]. One possibility for silicon transport is a diffusion-based mechanism, primarily from the cell body along the seta. Such an intracellular diffusion process can rely on high intracellular Si concentrations that were observed in other species[6,34,35], and can supply the needed Si at the seta tip. An alternative mechanism is transport by Si-containing vesicles. However, only occasionally vesicles were observed inside forming setae, and their interior lacked distinct electron-dense material that could be a dense Si-phase. Therefore, while vesicular transport cannot be ruled out, we currently consider a diffusion-based mechanism more likely. Regardless of the exact mechanism, our results suggest an intracellular Si-transport system that is under the control of the cell. Therefore even though seta formation occurs outside the cell membrane the process is still under a strict cellular control, from the confinement of the silicification volume, to the possible involvement of macromolecules, and the supply of the needed Si building blocks.

To conclude, even though it is well established that SDVs and intracellular processes are pivotal for diatom silicification, this work proposes an alternative mechanism, which is extracellular silicification. The simple cellular organization at the growing tip of the *C. tenuissimus* seta, together with the native-state structural information acquired by cryoET, have facilitated the elucidation of a silicification process that is regulated by the cell machinery, but that occurs outside the cell membrane. Similar processes might underlie many of the elongated and intricate silicified ornaments characteristic of diatom species that were hard to reconcile with intracellular formation.

general, and our data sets specifically, is sufficient to detect lipid bilayer membranes, for example, the cell membrane. Therefore, the fact that we do not observe any membrane-bound organelle that contains dense silica structures within actively silicifying setae strongly suggests that such silica deposition vesicles are not involved in the process. Based on the absence of any intracellular vesicle in which the seta forms, we deduce that this is an extracellular process, even though we are aware that a complete proof will need a comprehensive description of the process in action. Being a process that occurs in the extracellular environment does not mean it is happening spontaneously in bulk seawater without cellular regulation. The organic coat that surrounds the seta creates a confined nanoscale environment, which can be chemically very similar to the conditions inside an SDV. It is plausible that the cell membrane transfers the needed organic and inorganic building blocks into this extracellular space similarly to the transport mechanisms into an SDV. Finally, the observations that under silicification-limiting conditions seta development stops before SDV-mediated processes suggest that the regulation of the extracellular process is different, at least in some aspects, from that of the SDV.

This conclusion solves the inherent difficulties of the SDV paradigm for such long and continuous elements, whose exocytosis would involve massive membrane recycling and cellular coordination (Fig. 6a). An extracellular process easily accounts for

## Methods

**Chaetoceros tenuissimus culturing.** *C. tenuissimus* cultures were grown in natural seawater collected from the Mediterranean sea, filtered and sterilized. Before use, nutrients were supplemented according to f/2 recipe and enriched with Si. Cultures were maintained at 18 °C in 16/8 h light/dark cycles. For culturing at low Si concentrations a *C. tenuissimus* culture was centrifuged at 2000 × *g* for 5 min and resuspended in Si-free artificial seawater or natural Mediterranean seawater that contains ~2 μM Si; this step was repeated three times. 1 ml of culture at a concentration of 550,000 cells/ml was transferred into semipermeable plastic containers, 'thin-cert' (12 well 'thin cert', pore size 1 μm, Greiner bio-one), which floated on 400 ml reservoir containing artificial or Mediterranean seawater supplemented with Si to yield 330 μM, 10 μM, 5 μM, or 2.5 μM Si. Cell concentrations inside the 'thin cert' were measured once a week using a Z2 Coulter counter.

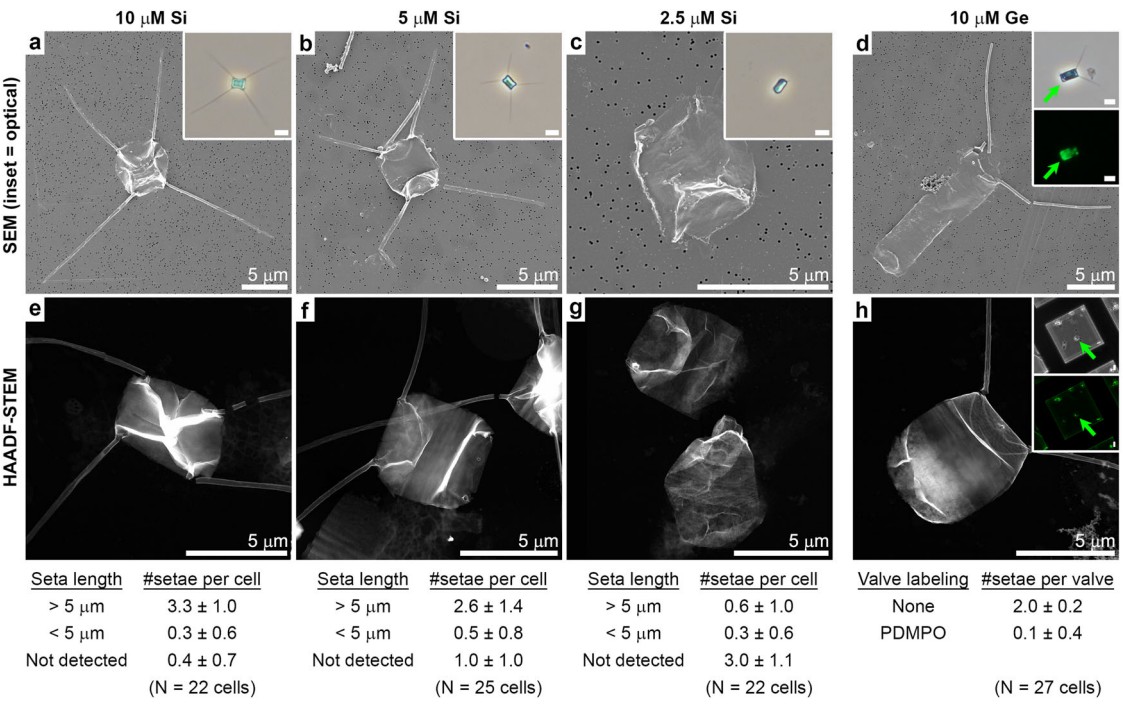

**Fig. 5 The influence of Si concentration and Si inhibitor in the growth medium on seta formation. a–d** SEM images of cells that were blotted dry for imaging, the insets show live cells. **e–h** HAADF-STEM images of washed and dried cell walls. The contrast in these images arise from the density of the material and is more affected by the folding of the cell walls then from actual variable densities of the material. Insets in **d** and **h** show optical microscopy images of the same cells as in the main panels. The fluorescence of PDMPO, which was added with Ge to the growth medium, allowed the identification of specific valves (indicated with green arrows) that formed in the presence of Ge. These valves do not have setae. For each experimental condition, a statistical summary of the average number and standard deviation of setae is presented below the images. In the different Si concentrations (**a–c**, **e–g**), the number of setae declines from a value that is close to the ideal four setae per cell to almost no setae at all. In the Ge-treated culture (**d**, **h**), the valves that are labeled with PDMPO, thus the ones that formed in the presence of Ge, have almost no setae, while the unlabeled valves that form prior to Ge addition have the expected two setae per valve.

Constant silicon content in the reservoir solutions was ensured by measuring silicon concentration once a week using a silicate test kit following the manufacturer's instructions (Merck Millipore, USA). The cells were diluted, once a week, using the appropriate medium.

For testing the influence of Ge, cultures growing in artificial seawater that contains 330 μM Si for a week were used. These cultures were diluted to 650,000 cells/ml and supplemented with 10 μM GeO$_2$ (stock solution of 100 mM GeO$_2$ (Sigma) dissolved in 32 mM NaOH) and 1 μM PDMPO (LysoSensorTM Yellow/Blue DND-160, Molecular Probes). These cultures were grown for a week.

**Synchronizing *C. tenuissimus* cell cycle.** *C. tenuissimus* cultures were diluted (10x) into fresh medium every 48 h, repeatedly, to maintain the culture in an exponential growth phase. To induce Si starvation, 10 ml aliquots of culture were centrifuged at 2000 × *g* for 5 min and resuspended in Si-free artificial seawater; this step was repeated three times. The cultures were then maintained for 12 h in a Si-free medium to arrest the cell cycle. After 12 h of Si starvation, Si was replenished to 330 μM and the cultures were placed on a shaker in continuous light to resume the cell cycle. To monitor the mineralization of new cell-wall components (valves, girdle bands and setae), 10 μl of 1 mM PDMPO (LysoSensorTM Yellow/Blue DND-160, Molecular Probes) was added to each 10 ml culture. Starting 4 h after the addition of Si, and at regular time intervals, cultures were imaged with a fluorescent microscope in order to evaluate silica formation and its synchronization. In most cultures, 30–50% of the cells were at the stage of seta formation after 6 h.

**Sample preparation for room temperature electron microscopy.** A *C. tenuissimus* culture was centrifuged at 2000 × *g* for 5 min and washed with MilliQ-grade water to remove salts from the sample. This was repeated twice and then cells were re-suspended in ethanol. For a quantitative STEM analysis of the valve Si density, the washed cells were re-suspended in 0.06% sodium hypochlorite and incubated at 4 °C for 24 h, with agitation. Then the solution was replaced by a new 0.06% sodium hypochlorite solution and the cells were incubated for an additional 24 h at 4 °C, with agitation. Cells were washed in MilliQ-grade water. For sample mounting, 7 μl of the sample was pipetted onto Formvar, lacey carbon, or carbon coated copper grids for TEM, or track-etched membranes for SEM.

**Sample preparation for cryoTEM tomography.** First, 5 μl of a synchronized *C. tenuissimus* culture was mixed with 10 nm gold nanoparticles, which served as fiducial markers[36]. The sample was pipetted onto freshly glow-discharged Quantifoil S7/2 grids coated with a 2–3 nm carbon film, and then blotted and plunged into liquid ethane with a Leica EM-GP plunge-freezing device. Frozen grids were kept in liquid N$_2$ dewar until used.

**Room temperature TEM.** Images of dried samples were taken on a Tecnai T12 microscope (Thermo Fisher Scientific) with a Gatan OneView camera. For HAADF-STEM imaging a Tecnai G2 F-20 scanning transmission electron microscope operating at 200 kV nanoprobe mode was used.

**cryoET.** Cryo-electron tomography data was collected on a Titan Krios TEM (Thermo Fisher Scientific), equipped with a Falcon III direct electron detector and a BioQuantum energy filter with a K3 direct electron detector (Gatan Inc.). Data sets were collected at 300 kV either with the Falcon III (integrated mode) with a Volta phase plate and target defocus of ~1 μm underfocus using the manufacturer operation package, or on the K3 camera (counting mode) without the phase plate, and target defocus of ~3 μm underfocus, using SerialEM software. Total dose per tilt series was limited to ~100 e$^-$/Å$^2$. Tomogram tilt schemes were bi-directional, from either 0° or 20° as starting points, sweeping through ±60° at 2° steps.

**Data analysis and 3D representation.** The cryoTEM tilt series image data sets were aligned, CTF-corrected when applicable, and reconstructed using the IMOD 4.9.12 software package[37]. Segmentation and 3D representation of the reconstructed tomographic data was done using Amira® software (Thermo Scientific). Data segmentation was performed based on contrast variations following the unique shape and structure of each component. Modeling of the seta 3D structure and the construction of a model for seta elongation were undertaken using educational license Autodesk Maya 2018. The 3D model was built by fitting structures and their relative position to the reconstructed cryoTEM tomography data.

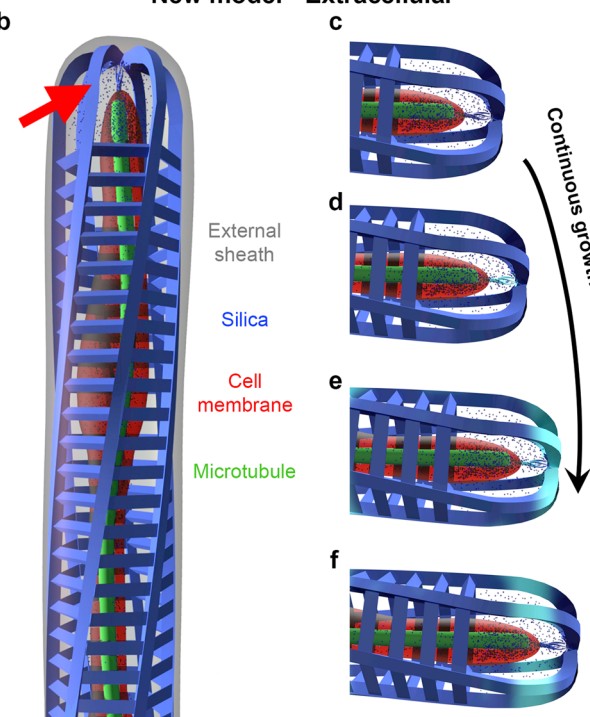

**Fig. 6 Schematic representations of SDV-mediated and non SDV-mediated models for seta formation. a** An SDV-mediated process that includes an SDV membrane surrounding the forming silica, and a dedicated cellular complex that is responsible for elongation of the seta (based on the works of J. Pickett-Heaps). Only after maturation the silica is exocytosed and become exterior to the cell membrane. **b** The new model based on the components identified by cryoET, with the addition of Si building blocks as small spheres, and the putative organic scaffolds that emanate from the far end of the cell cytoplasm, indicated with a red arrow. **c–f** Steps in the seta elongation process. A newly formed section of the organic scaffold (cyan in **d–f**) silicifies upon radial expansion, and then is pushed down along the seta. The organic cover was removed for clarity.

**Reporting summary**. Further information on research design is available in the Nature Research Reporting Summary linked to this article.

## Data availability

The data that support the findings of the study are available from the corresponding author upon reasonable request.

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

## Acknowledgements
We thank Yuji Tomaru and Kim Thamatrakoln for isolating and providing the algal cultures, and Eyal Shimoni for his help with the HAADF-STEM measurements. This project has received funding from the European Research Council (ERC) under the European Union's Horizon 2020 research and innovation programme (grant agreement no. 848339), and the Irving and Cherna Moskowitz Center for Nano and BioNano Imaging (Weizmann Institute of Science).

## Author contributions
B.M., L.A. and S.G.W. performed the experiments. B.M., L.A., S.G.W. and N.V. analyzed the results. N.V. constructed the 3D model. A.G supervised the research and wrote the paper with the assistance of all the authors.

## Competing interests
The authors declare no competing interests.
