## [Peer Review File · Nature Communications]

Reviewers' Comments:

Reviewer #1 (Remarks to the Author):

The manuscript by Mayzel et al. deals with the mechanism of seta formation in the diatom genus *Chaetoceros*. This biomineralization process is extremely fast, producing ~10 micrometer sized, nanopatterned silica whiskers within less ~20 minutes. Using a cell cycle synchronization procedure in combination with cryo-TEM analysis, the authors have imaged under near-native conditions both growing setae and setae that, presumably, have reached their full length. The images reveal in unprecedented detail the protoplast structure of growing and mature setae and the presence of an external organic layer that covers the entire distal surface of the silica. A striking result of the analysis is the apparent absence of silica deposition vesicles (SDVs) in the protoplasm of growing setae. This result is the sole evidence for the authors' claim that seta formation in *Chaetoceros* proceeds through extracellular rather than SDV bound intracellular mineralization.

There are certainly good reasons to believe that silica formation in diatoms can also happen extracellularly as it occurs in sponges and land plants. However, in neither of these two groups does the silica exhibit such regular silica nanopatterns as is observed in the diatom setae. An extracellular silica biomineralization process with such a high level of morphological control would thus be unique. Unfortunately, the model for seta formation proposed by Mayzel and co-workers does not provide an explanation for the patterning process. It does not include the formation of the cross connections that link the longitudinal silica fibers. The model seems to imply that the longitudinal fibers are under tension, because they are bent in a dome-like fashion at the tip of the setae and then become oriented longitudinally as the seta grows. What is the role in this process of the "putative organic scaffolds that emanate from the far end of the cell cytoplasm"? Is this scaffold connected to the external organic layer that surrounds the silica? What is the role of the external organic layer in the mineralization process? Obviously, one cannot expect hypotheses that address each of these key questions, but Mayzel and co-workers do not put address any of them.

It is indeed very surprising that no SDVs could be seen in any of their samples, yet there are many spherical vesicles present near the tip of the growing setae. They lack solid silica, but are they filled with a soluble Si precursor? This could be addressed by EDXA or EELS, but I am not sure whether any of the two are compatible with cryoTEM. Could the vesicles undergo very rapid morphogenesis and exocytosis of solid silica fragments, which then assemble extracellularly? Such a strategy is used by choanoflagellates to build up their extracellular baskets made of silica fibers that are intracellularly produced in individual SDVs.

The weakest aspect of the paper is the lack of positive evidence for extracellular mineralization. This process should be extremely sensitive to the silicic acid concentration in the medium. One would expect that seta formation is slowed down and/or their silica content is reduced, when silicic acid concentration is reduced to silicic acid levels that still support cell division, but are substantially lower than the standard concentration in the medium. Do setae become more heavily silicified, when the silicic acid concentration in the medium is substantially increased? Additionally, a fluorescently tagged, non-membrane permeable silicic acid molecule (e.g. prepared from aminopropyl trimethoxysilane modified with FITC) could be added to the medium during seta growth. If the mineralization process happens extracellularly the seta should become fluorescein labeled. Neither of these two experiments is simple, because appropriate controls have to be made and conditions established to rule out that (i) SDV mediated mineralization in *Chaetoceros* is affected by extracellular silicic acid concentrations, and (ii) the fluorescently tagged silicic acid does not label extracellular silica or the external organic layer. Nevertheless, in the absence of any positive evidence that could stem from the suggested experiments above (or more elegant ones that the authors may be able to devise), I regard the claim premature that extracellular silica formation in diatoms has been discovered.

Reviewer #2 (Remarks to the Author):

The manuscript by Mayzel et al present a set of very interesting observations that point to a role for extracellular silicification in the formation of diatom setae. It has long been proposed that these elongated appendages could not be a product of internal silicification through the normal process involving an intracellular silica deposition vesicle. The detailed EM and fluorescent microscopy analysis presented provides good evidence, including the presence of an external sheath surrounding the silicified structure and a bilayer within that along with the indication that silica deposition occurs at the tip of the growing seta. This system could potentially provide an accessible model for better understanding of the molecular and chemical basis of silica deposition. The hypothesis is based on structural description and could be strengthened by the addition of some further analyses or experimentation.

The hypothesis that the diffusion of Si occurs from seawater concentrations across the organic sheath with the creation of a potent chemical environment poses a number of conceptual problems. It is difficult to see how a favourable potent environment can be maintained that must involve an impermeable covering while still allowing diffusional transport of Si inwards. The authors argue that diffusion of Si along the seta would be unlikely since it would require a large concentration gradient of Si. This argument needs to be supported by knowledge of the diffusion coefficient of soluble Si and concentration of Si in the cell body. Since it is argued that precipitation from seawater concentrations of Si is likely then maintaining a suitable concentration gradient may not represent such a constraint. Alternatively, Si could be transported along the seta length by other unidentified mechanisms. Clearly transport of other components of the seta (proteins, polysaccharide) occurs from where they are made in the cell body to the tip. Si could then be actively transported across the cell membrane to the external silicifying space by the action of silicon transporters, albeit operating in a different direction to normal Si transporters. Some of these possibilities are testable, for example by testing the effects of known Si transport inhibitors and by examining the dependence of seta silicification on external Si concentration, which should differ considerably from that of the frustule according to the hypothesis presented. Such relatively simple experiments would provide physiological evidence to support the hypothesis based currently solely on structural observations.

Fig 1 d and e: Is there a focus issue? Only the tip of the seta appears to be in focus in this image pair. There is a hint of labelling further back towards the cell body but this is obscured by fluorescence from the cell body. The authors should provide more convincing images to show labelling restricted to the tip. For example, confocal sections would ensure that all of the seta length was correctly in focus.

Reviewer #3 (Remarks to the Author):

REVIEW OF "EXTRACELLULAR SILICA FORMATION BY DIATOMS" BY GAL ET AL.

1. What are the major claims of the paper? Are the claims novel? If not, please identify the major papers that compromise novelty

The paper claims a new silicification process in diatoms that is extracellular and does not involve a silica deposition vesicle (SDV). It concerns more particularly the formation of the setae. Most studies on diatom silicification focus on valve formation, and all report an intracellular process with SDV involved. There is so far and to my knowledge, very little literature on setae formation. I do not know any other recent studies than the ones from Pickett Heap (1994 and 1998, cited in the paper) who proposed two alternative mechanism of setae extension, but both involving SDV and intracellular silicification.

2. Will the paper be of interest to others in the field? On a more subjective note, do you feel that the paper will influence thinking in the field?

The paper will be of great interest for the whole community of diatomist, biogeochemist, plankton

physiologist, ... to achieve a more complete understanding of silicification processes and associated fluxes. It may also ultimately provide new inspiration sources for the preparation of nanoscale and nanostructured silica-based materials for a host of applications in nanotechnology.

3. Is the work convincing, and if not, what further evidence would be required to strengthen the conclusions?

The main problem with the paper is that the claims are not very convincing.

1) The PDMPO fluorescence image shows a labeled setae tip (Fig 2 d, e), but the image is blurred and shows only one cell. It would be nice to capture several relevant images of several cells at regular intervals during the 20 minutes of setae formation to provide more information on the process and to show the evolution of the labelling along the setae from the newly formed cell wall. The fact that setae are growing from the tip is not new, but the way of showing it with PDMPO fluorescence would have been innovative if pictures were of better quality.

2) It is rather awkward to develop the new theory (no SDV, extracellular silicification process) based only on things (SDV) authors did not see.

L194-198: "... the fact that we do not observe any silica deposition vesicles within actively silicifying setae means that they are not involved in the process. ... based on which we deduce that this is an extracellular process."

The results are over-interpreted and there are far too many hypotheses for the external silicification process, although interesting, to be plausible. The formation of transversal elements, which connect the longer rods, is not explain and is not compatible with the process proposed. A pedagogical effort to explain the extension of the setae is made with the 3D representation of the reconstructed tomographic data but the process is still obscure. The additional material that is a video of the reconstructed images does not bring any additional information than the schematic representation (Fig. 5)

3) The extracellular source of silicic acid proposed, from seawater, has a concentration usually between 2 and 10 μM . Silica precipitation from so low concentrations is highly hypothetical even considering the potential role of the external sheath that covers the silica elements. The characterization of the proteins that constitute the external sheath could help to understand its role in silicification.

4. Are there other experiments that would strengthen the paper further?

I am not a specialist, but I would recommend to image several cells and to add PDMPO at regular time intervals to show the evolution of the labelling along the setae.

More pictures with the cryoTEM could provide more details on the formation of transversal elements and a more comprehensive explanation of the process of setae growth. Moreover, characterizing the structure of the external sheaths could help understanding its role in the silicification process (see Hildebrand et al 2018).

5. Are the claims appropriately discussed in the context of previous literature?

That's another major drawback of the paper. Results are not really discussed in the context of previous literature. They are not compare with previously proposed mechanisms of setae extension (as the ones described by Pickett-Heaps et al (1998), for example, who says that extension at the setae tip is generated by a complex of cell organelles (SDV and microtubules). They are neither compared to studies proposing extracellular silicification processes in other organisms like in sponges for example (Wang et al 2011, others?) (some other comments are directly annotated in the manuscript)

6. If the manuscript is unacceptable in its present form, does the study seem sufficiently promising that the authors should be encouraged to consider a resubmission in the future?

The manuscript in the present form is unacceptable, but authors should consider resubmission if they can show data that are more convincing and provide a comprehensive explanation of the external silicification process that includes the formation of the silica transversal elements.

Response to Reviewers

We are excited to submit a new version of the manuscript that contains more evidence to support our conclusion that silica formation in diatoms is possible outside the cell membrane. As the reviewers mentioned, this finding by itself is transformative to the field of diatom silicification.

First, we would like to address generally some considerations of our study that were raised by all three expert reviewers: It is clear that the biggest challenge in substantiating the conclusion that the setae are silicified outside the cytoplasm is to prove that something (an SDV) does not exist. Our strategy to deal with this complication is four-fold:

1. We image seta formation with good enough preservation and resolution such that if there was an SDV it should have been detected. We understand that this logic is unusual, as noted by the reviewers, but nevertheless, completely valid. We want to stress that in the case of seta formation, an SDV-mediated process is not more likely or simple, it will just conform to the main paradigm. Therefore, we feel confident that our careful observations using cryoET on seta formation demonstrate that the silica phase is present only outside the cell.

2. The structural properties of SDVs in general are still a scarcely addressed question, with most of the available information collected by conventional electron microscopy. In our group, we use advanced electron microscopy approaches to investigate structural properties of the SDV. Our unpublished results show that with advanced sample preparation protocols an SDV is easy to spot when it exists. For example, the image below shows a slice in an electron tomogram of a high-pressure-frozen, freeze-substituted and resin embedded diatom cell, and the SDV is clearly visible:

Therefore, the fact that we do not observe an SDV in the seta forming cells demonstrates a process that does not involve this organelle.

3. We are aware that it was much better if we could provide a detailed alternative for the formation mechanism of the seta, which will explain silicon transport, concentration gradients, biochemistry of the organic scaffold, and the morphogenetic regulation of the helical rods. However, such detailed understanding is not available even for the better investigated SDV-mediated silicification. Therefore, the focus of this research is only the extracellular nature of the formation process and our schematic model is intended only as a simplistic working hypothesis to inspire future research.

4. Lastly, and most important, the suggestions of the reviewers triggered further experiments that provided the needed positive evidence for the extracellular mechanism of seta silicification. Even though several experimental directions were not fruitful (detailed below), manipulations of growth media of the cells allowed us to provide the needed proof that the formation of the setae responds differentially to external conditions (the new Fig. 5). With these added results, which are described in detail below, we hope that the reviewers will be convinced that we have enough support for our conclusions.

Below is our detailed response (in blue) to the reviewer report. In addition, a revised manuscript file, where all the changes are marked in red, was uploaded to the submission system.

Reviewer #1 (Remarks to the Author):

The manuscript by Mayzel et al. deals with the mechanism of seta formation in the diatom genus *Chaetoceros*. This biomineralization process is extremely fast, producing ~10 micrometer sized, nanopatterned silica whiskers within less ~20 minutes. Using a cell cycle synchronization procedure in combination with cryo-TEM analysis, the authors have imaged under near-native conditions both growing setae and setae that, presumably, have reached their full length. The images reveals in unprecedented detail the protoplast structure of growing and mature setae and the presence of an external organic layer that covers the entire distal surface of the silica. A striking result of the analysis is the apparent absence of silica deposition vesicles (SDVs) in the protoplasm of growing setae. This result is the sole evidence for the authors' claim that setae formation in *Chaetoceros* proceeds through extracellular rather than SDV bound intracellular mineralization.

1) We agree that the cryoET results are the sole **direct** evidence for the extracellular formation. In the revised version we added several **indirect** evidence (new Fig. 5, detailed below) that support a distinct mechanism for seta formation that does not involve an SDV. The manuscript title was revised to emphasize this and now reads: "Structural evidence for extracellular silica formation by diatoms".

There is certainly good reasons to believe that silica formation in diatoms can also happen extracellularly as it occurs in sponges and land plants. However, in neither of these two groups does the silica exhibit such regular silica nanopatterns as is observed in the diatom setae. An extracellular silica biomineralization process with such high level of morphological control would thus be unique.

Unfortunately, the model for setae formation proposed by Mayzel and co-workers does not provide an explanation for the patterning process. It does not include the formation of the cross connections that link the longitudinal silica fibers. The model seems to imply that the longitudinal fibers are under tension, because they are bend in a dome-like fashion at the tip of the setae and then become oriented in a longitudinally as the seta grows. What is the role in this process of the "putative organic scaffolds that emanate from the far end of the cell cytoplasm"? Is this scaffold connected to the external organic layer that surrounds the silica? What is the role of the external organic layer in the mineralization process? Obviously, one cannot expect hypotheses that address each of these key questions, but Mayzel and co-workers do not put address any of them.

2) Indeed, our model does not address many of the details of the intricate morphology of the seta, but it offers a framework for future work. For example, our findings can provide insights into two important aspects of seta formation outlined by the reviewer: the external organic layer separates the forming silica from the bulk seawater, thus providing a preferred chemical environment that can be similar to the conditions inside an SDV. Second, our suggestion of pre-stressing the longitudinal fibers can explain the structural rigidity of the setae, as it was long questioned how such elongated structures sustain the local forces that a cell can encounter.

We revised these points in the discussion:

Lines 258-261 “However, being a process that occurs in the extracellular environment does not mean it is happening spontaneously in bulk seawater without cellular regulation. The organic coat that surrounds the seta creates a confined nanoscale environment, which can be chemically very similar to the conditions inside an SDV.”

Lines 277-282: “An interesting consequence of such a process is that the silica rods are first mineralized in a direction perpendicular to the seta length and only later are aligned with it. This will create internal stress in the material, which may stabilize it, thus explaining why these delicate long structures do not bend even under large forces (such as the centrifugation and drying needed for sample preparation), and maintain their straight orientation³⁰.”

It is indeed very surprising that no SDVs could be seen in any of their samples, yet there are many spherical vesicles present near the tip of the growing setae. They lack solid silica, but are they filled with a soluble Si precursor? This could be addressed by EDXA or EELS, but I am not sure whether any of the two are compatible with cryoTEM. Could the vesicles undergo very rapid morphogenesis and exocytosis of solid silica fragments, which then assemble extracellularly? Such a strategy is used by choanoflagellates to build up their extracellular baskets made of silica fibers that are intracellularly produced in individual SDVs.

3) We considered using cryoEDX\EELS to investigate Si content of the seta. Unfortunately, the external silica coverage of the seta always exceeds and masks the signal coming from the cytoplasm and we could not de-convolve the two. In the revised version we added data from more tomograms (Fig. S1) showing that the small vesicles are not very abundant in the data sets (since this is a rapid process it would have been expected if this was a major pathway of transport). Even if such vesicles do contribute to silica formation they do not contain a dense silica phase and are not an SDV-like compartment, thus not directly affecting the goal of the paper.

Related to this point is the revision we made to the paragraph about the possible Si transport mechanism (in response also to other comments). It is now in lines 308-314: “An interesting open question regarding this process is the source of the silicon building blocks used to form the seta. As we only occasionally observed vesicles inside forming seta, and as, in general, vesicular transport of silica building blocks is not established in diatoms^{5,6,31,32}, we hypothesize that silicon is transported via a diffusion-based mechanism, primarily from the cell body along the seta. Such an intracellular diffusion process can rely on high intracellular Si concentrations that were observed in other species,^{6,33,34} and can supply the needed Si at the seta tip.”

The weakest aspect of the paper is the lack of positive evidence for extracellular mineralization. This

process should be extremely sensitive to the silicic acid concentration in the medium. One would expect that seta formation is slowed down and/or their silica content is reduced, when silicic acid concentration is reduced to silicic acid levels that still support cell division, but are substantially lower than the standard concentration in the medium. Do setae become more heavily silicified, when the silicic acid concentration in the medium is substantially increased?

4) As noted above, we took special efforts in addressing this point and collecting more data to support our results. We planned and performed dedicated experiments where we grew cultures at varying Si concentrations. In order to keep a very low, but constant, silicon concentration, we grew the cells in semi-permeable containers that were situated in contact with a large reservoir of growth medium so the silicification process will not further deplete the Si concentration. These experiments showed that at very low Si concentrations of ~2.5 μM the cells reproduce, form their cell walls, but not their setae. This result gives a positive indication that seta formation proceeds via a different mechanism from valve formation. In addition, we grew the cells with germanium and identified intermediate concentrations that inhibited specifically the formation of the setae. Again, this result positively shows differences between seta formation and the classical SDV-mediated processes.

These data are added to the text in the new Fig. 5 and the adjacent text in lines 181-236.

Additionally, a fluorescently tagged, non-membrane permeable silicic acid molecule (e.g. prepared from aminopropyl trimethoxysilane modified with FITC) could be added to the medium during seta growth. If the mineralization process happens extracellularly the seta should become fluorescein labeled. Neither of these two experiments is simple, because appropriate controls have to be made and conditions established to rule out that (i) SDV mediated mineralization in *Chaetoceros* is affected by extracellular silicic acid concentrations, and (ii) the fluorescently tagged silicic acid does not label extracellular silica or the external organic layer. Nevertheless, in the absence of any positive evidence that could stem from the suggested experiments above (or more elegant ones that the authors may be able to devise), I regard the claim premature that extracellular silica formation in diatoms has been discovered.

5) The proposed experiment with a non-membrane permeable silicon precursor makes a lot of sense but is likely to be difficult to calibrate so we opted for the first option of varying the Si concentration of the medium, as detailed in the previous point. Additionally, the new results that show a lag time between changing the external conditions and the effect on the setae (we address this in detail in the response to point (2) of Reviewer #2), suggest that the main Si transport pathway is through the cell. Such pathway will make the incorporation of external Si species less likely.

In addition to the two suggestions of this reviewer, we attempted two other experiments that did not yield informative results. We briefly outline these here just to express how elusive is the study of seta formation:

- 1) We tried to compare trace element incorporation into the seta-silica vs. valve-silica. We added both Al and Ge to the growth medium to explore the hypothesis that the extracellular process will result in higher amounts of impurities (which in the case of an SDV will be screened by the selectivity of the intracellular process). However, Al was observed to adhere to all silicified cell wall components, and Ge inhibited the formation of the setae (as shown in Fig. 5).

- 2) We tried to use correlative PDMPPO fluorescence and advanced sample preparation procedures for room temperature electron microscopy that will enable us to image also the cell body and the base of the setae. However, this method proved to be too low-throughput to serve as a robust method.

Reviewer #2 (Remarks to the Author):

The manuscript by Mayzel et al present a set of very interesting observations that point to a role for extracellular silicification in the formation of diatom setae. It has long been proposed that these elongated appendages could not be a product of internal silicification through the normal process involving an intracellular silica deposition vesicle. The detailed EM and fluorescent microscopy analysis presented provides good evidence, including the presence of an external sheath surrounding the silicified structure and a bilayer within that along with the indication that silica deposition occurs at the tip of the growing seta. This system could potentially provide an accessible model for better understanding of the molecular and chemical basis of silica deposition. The hypothesis is based on structural description and could be strengthened by the addition of some further analyses or experimentation.

- 1) We thank the reviewer for this opinion and agree that strengthening the hypothesis improved significantly the manuscript (new data of Fig. 5, which is detailed in point (4) of the response to Reviewer #1).

The hypothesis that the diffusion of Si occurs from seawater concentrations across the organic sheath with the creation of a potent chemical environment poses a number of conceptual problems. It is difficult to see how a favourable potent environment can be maintained that must involve an impermeable covering while still allowing diffusional transport of Si inwards. The authors argue that diffusion of Si along the seta would be unlikely since it would require a large concentration gradient of Si. This argument needs to be supported by knowledge of the diffusion coefficient of soluble Si and concentration of Si in the cell body. Since it is argued that precipitation from seawater concentrations of Si is likely then maintaining a suitable concentration gradient may not represent such a constraint. Alternatively, Si could be transported along the seta length by other unidentified mechanisms. Clearly transport of other components of the seta (proteins, polysaccharide) occurs from where they are made in the cell body to the tip. Si could then be actively transported across the cell membrane to the external silicifying space by the action of silicon transporters, albeit operating in a different direction to normal Si transporters. Some of these possibilities are testable, for example by testing the effects of known Si transport inhibitors and by examining the dependence of seta silicification on external Si concentration, which should differ considerably from that of the frustule according to the hypothesis presented. Such relatively simple experiments would provide physiological evidence to support the hypothesis based currently solely on structural observations.

2) We agree that the previous discussion of Si transport was not based on evidence, and the new data indeed pointed to different directions, which are reflected in the new version. We performed the experiments suggested by this reviewer (and by the other reviewers), namely, growing the cells with varying Si concentrations and in the presence of Ge, a known silicification inhibitor. The presence of germanium, as well as low Si concentration resulted in cells that silicify but did not form setae, as described in the new Fig. 5 and the adjacent text in lines 181-236. Importantly, short-term exposure to low Si did not immediately inhibit seta development and it took several generations to observe the total effect of this stress. We interpret these results as the outcome of an intracellular Si transport mechanism, in which the cell concentrates intracellular Si pool that is extruded to the confined volume at the seta exterior where polymerization takes place. We agree with the reviewer comment that this might be similar to transport across the SDV membrane, but here it is the cytoplasm membrane. A rough calculation with a diffusion coefficient for a small molecule of $\sim 10^{-6} \text{ cm}^2 \text{ s}^{-1}$ gives diffusion times in the order of seconds across the 10 microns of the seta. Therefore, both diffusion and vesicle transport can facilitate this flux of building blocks along the seta.

We entirely revised the paragraph about Si transport in the discussion, which now is: Lines 308-314: “An interesting open question regarding this process is the source of the silicon building blocks used to form the seta. As we only occasionally observed vesicles inside forming seta, and as, in general, vesicular transport of silica building blocks is not established in diatoms^{5,6,31,32}, we hypothesize that silicon is transported via a diffusion-based mechanism, primarily from the cell body along the seta. Such an intracellular diffusion process can rely on high intracellular Si concentrations that were observed in other species,^{6,33,34} and can supply the needed Si at the seta tip.”

Fig 1 d and e: Is there a focus issue? Only the tip of the seta appears to be in focus in this image pair. There is a hint of labelling further back towards the cell body but this is obscured by fluorescence from the cell body. The authors should provide more convincing images to show labelling restricted to the tip. For example, confocal sections would ensure that all of the seta length was correctly in focus.

3) Yes, focus is a major issue for live-cell imaging of the extremely narrow setae. Unfortunately, also photo-bleaching is a major concern and therefore confocal microscopy proved to be incompatible with this type of cellular structures. We invested substantial time in collecting more images but we could not improve the quality of this image significantly. As it is established in the literature that the seta grows from the tip we did not pursue this experimental problem further and only revised the text to mention how rare it is to capture this type of images. Lines 91-95: “It was previously shown for different *Chaetoceros* species that the setae elongate from their tip^{15,24}. In accordance with such tip growth process, we could sometime observe the furthestmost part of the seta stained with PDMPO when the dye was added for a couple of minutes to a synchronized culture at the stage of seta formation (Fig. 2d,e).”

More details about the experimental problems in the response to point (1) of Reviewer #3.

Reviewer #3 (Remarks to the Author):

1. What are the major claims of the paper? Are the claims novel? If not, please identify the major papers that compromise novelty

The paper claims a new silicification process in diatoms that is extracellular and does not involve a silica deposition vesicle (SDV). It concerns more particularly the formation of the setae. Most studies on diatom silicification focus on valve formation, and all report an intracellular process with SDV involved. There is so far and to my knowledge, very little literature on setae formation. I do not know any other recent studies than the ones from Pickett Heap (1994 and 1998, cited in the paper) who proposed two alternative mechanism of setae extension, but both involving SDV and intracellular silicification.

2. Will the paper be of interest to others in the field? On a more subjective note, do you feel that the paper will influence thinking in the field?

The paper will be of great interest for the whole community of diatomist, biogeochemist, plankton physiologist, ... to achieve a more complete understanding of silicification processes and associated fluxes. It may also ultimately provide new inspiration sources for the preparation of nanoscale and nanostructured silica-based materials for a host of applications in nanotechnology.

3. Is the work convincing, and if not, what further evidence would be required to strengthen the conclusions?

The main problem with the paper is that the claims are not very convincing.

1) The PDMPO fluorescence image shows a labeled setae tip (Fig 2 d, e), but the image is blurred and shows only one cell. It would be nice to capture several relevant images of several cells at regular intervals during the 20 minutes of setae formation to provide more information on the process and to show the evolution of the labelling along the setae from the newly formed cell wall. The fact that setae are growing from the tip is not new, but the way of showing it with PDMPO fluorescence would have been innovative if pictures were of better quality.

1) The main goal of the PDMPO labeling was to synchronize the cultures for the cryoET work, and the observation of the labeled tip was only a nice byproduct of the experiment. Following the reviewer suggestions we tried to expand these experiments so we can get dynamic information on seta growth. Unfortunately, this process proved too delicate and variable to yield reproducible results. There are too many complications, one is that the synchronization is partial (similar to other species). Second, PDMPO staining of the setae is very weak and prone to photobleaching so we cannot follow the growth of a single seta, and third, the incorporation of PDMPO into the silica is a physiological process that is not well understood. In some cells the dye can be seen in a matter of minutes within new silica and in others it requires hours. We also think that thickening of the seta can occur along its entire length while it is still elongating, so the initial labeling of the tip can be very fast masked by further incorporation into the entire seta.

In summary, we could not expand on the process of seta formation using PDMPO labeling. We revised the text to make the limitations of these experiments clear. Lines 91-95: “It was previously shown for different *Chaetoceros* species that the setae elongate from their tip^{15,24}. In accordance with such tip growth process, we could sometime observe the furthest part of the seta stained with PDMPO when the dye was added for a couple of minutes to a synchronized culture at the stage of seta formation (Fig. 2d,e).”

2) It is rather awkward to develop the new theory (no SDV, extracellular silicification process) based only on things (SDV) authors did not see.

L194-198: “... the fact that we do not observe any silica deposition vesicles within actively silicifying setae means that they are not involved in the process. ... based on which we deduce that this is an extracellular process.”

The results are over-interpreted and there are far too many hypotheses for the external silicification process, although interesting, to be plausible. The formation of transversal elements, which connect the longer rods, is not explain and is not compatible with the process proposed. A pedagogical effort to explain the extension of the setae is made with the 3D representation of the reconstructed tomographic data but the process is still obscure. The additional material that is a video of the reconstructed images does not bring any additional information than the schematic representation (Fig. 5)

2) Indeed our main finding is that an SDV is not involved in seta formation. We address this issue at the beginning of this letter. We hope that the added results that support the notion of a distinct silicification mechanisms for the setae convince the reviewer that this is by itself an interesting observation that is changing fundamental views about diatom silica. The formation of the helical rods and transversal elements is still a mystery that we do not claim to solve. Please note that what is regulating these elements, as well as many other structural elements of diatom silica, is an open area for research regardless if it is formed within an SDV or outside the cytoplasm.

3) The extracellular source of silicic acid proposed, from seawater, has a concentration usually between 2 and 10 μM . Silica precipitation from so low concentrations is highly hypothetical even considering the potential role of the external sheath that covers the silica elements. The characterization of the proteins that constitute the external sheath could help to understand its role in silicification.

3) The new data have modified our view on the Si transport pathway (as explained in detail in our response to point (2) of Reviewer #2). We propose that the silicon source is the intracellular pool that is being transported to the confined environment between the cell membrane and the organic coat, which can have chemical composition very different from seawater. We did initial attempts to extract and isolate this organic coat but currently we do not have any handle on how to separate it from other organic constituents of the cell and cell-wall.

4. Are there other experiments that would strengthen the paper further?

I am not a specialist, but I would recommend to image several cells and to add PDMPO at regular time intervals to show the evolution of the labelling along the setae.

4) Please see our response in point (1) regarding PDMPO experiments.

More pictures with the cryoTEM could provide more details on the formation of transversal elements and a more comprehensive explanation of the process of setae growth. Moreover, characterizing the structure of the external sheaths could help understanding its role in the silicification process (see Hildebrand et al 2018).

5) We have now included in the new Fig. S1 slices that represent all the tomograms acquired for the various setae. We think this will help the reader to judge what the typical features of the setae are and which aspects are sample-specific.

5. Are the claims appropriately discussed in the context of previous literature?

That's another major drawback of the paper. Results are not really discussed in the context of previous literature. They are not compared with previously proposed mechanisms of setae extension (as the ones described by Pickett-Heaps et al (1998), for example, who says that extension at the setae tip is generated by a complex of cell organelles (SDV and microtubules). They are neither compared to studies proposing extracellular silicification processes in other organisms like in sponges for example (Wang et al 2011, others?)

6) We improved the relation to previous works within the manuscript:

a) in the introduction: Lines 36-38 "Although it is known that other organisms can form silica extracellularly^{10,11}, the current view on diatom silicification is that it is invariably an intracellular process^{2,3}.", Lines 47-49 "Such extensions challenge the concept of intracellular silicification, resembling the extremely long spicules formed by sponges in a process that may start intracellularly but is predominantly extracellular¹⁴."

b) In Fig. 6, a new panel was added that shows the previous suggestions for an SDV related seta formation. The Figure caption now reads: "Fig. 6. Schematic representations of SDV-mediated and non SDV-mediated models for seta formation. a) An SDV-mediated process that includes SDV membrane (in green) surrounding the forming silica (based on the works of J. Pickett-Heaps). b-f) The new model based on the components identified by cryoET..."

c) In the discussion: Line 273-274 "Such an insoluble organic scaffold has been identified in several diatom species and resembles the process of spicule formation by sponges^{14,19,27-29}."

(some other comments are directly annotated in the manuscript)

7) Following the comments in the annotated file the following modifications have been made"

a) There is a contradiction between: "the thin setae of this species do not allow any cellular organelles to penetrate the seta volume" and "the presence of an observable cellular element, a microtubule". Some membranous vesicles were also observed

What would be the minimum size of a setae that could include mitochondria, SDV,.. to participate in the setae formation? Lines 241-244 “First, the thin setae of this species do not allow large cellular organelles to penetrate the seta volume, leaving a bare-bone system, where silica forms at the seta tip in the presence of only a single observable cellular element, a microtubule, and occasionally small vesicles.”

b) Does this mean that the proposed process is species specific ? and occurs only in species that have very thin setae?, Would the process of extracellular silicification be specific to *C. tenuissimus* species only? Lines 249-251 “The common features between the described process and seta formation in other *Chaetoceros* species are yet to be explored.”

c) But it is not a problem if the setae is formed in successive steps Lines 267-269 “This conclusion solves the inherent difficulties of the SDV paradigm for such long and continuous elements, whose exocytosis should involve massive membrane recycling and cellular coordination (Fig. 6a).”

d) to which concentration? Lines 349-350 “After 12 hours of Si starvation, Si was replenished to 330 μM and the cultures were placed on a shaker in continuous light to resume the cell cycle.”

6. If the manuscript is unacceptable in its present form, does the study seem sufficiently promising that the authors should be encouraged to consider a resubmission in the future?

The manuscript in the present form is unacceptable, but authors should consider resubmission if they can show data that are more convincing and provide a comprehensive explanation of the external silicification process that includes the formation of the silica transversal elements.

8) We hope that the revised version and the added new experiments meet these requirements.

REVIEWER COMMENTS

Reviewer #1 (Remarks to the Author):

The revised title of the manuscript better reflects the current state of knowledge, because also the new data still do not provide clear-cut evidence for extracellular silica formation. The added experiment on seta formation in the presence of very low silicic acid concentration did not show "instantaneous" (i.e. during the next cell division) cessation of seta formation. Instead it took several generations before a reduction in the length and number of setae took effect. The authors therefore suggest that seta formation is fueled from intracellular silicic acid pools where silicic acid is concentrated, and then exported into the extracellular matrix where they believe the biomineralization of setae to take place. While this scenario seems feasible, it relies on using the general intracellular Si pool for seta formation. As a consequence, the experiment is unable to decouple Si use for biomineralization in the SDV from the use of Si for the proposed extracellular seta biomineralization. Therefore, the Si-limitation experiment can neither support nor disprove an extracellular biomineralization mechanism for seta formation.

Nevertheless, the excellent cryoTEM data are sufficient enough to seriously question the previous model for seta formation (as was put forward by Pickett-Heaps), and the authors provide a feasible alternative model for this process. By changing the title and also throughout the text the authors no longer claim to have PROVEN the existence of extracellular silica biomineralization in diatoms. Instead they claim, justifiably so, that their data is consistent with extracellular biomineralization. I regard this paper as very important, and recommend it to be published after the following additions and modifications have been made:

1. In the rebuttal letter the authors present an image from HP-frozen/freeze substituted diatom cell that contains silica filled SDVs. Apparently, the cell does not seem to be from the same diatom species that was used for the present study on seta development. The fact that a different fixation method and a different species was used is not really a proof for the ability to visualize SDVs in cryofixed cells of *C. tenuissimus*. Therefore, the authors should provide in the supplementary information images from cryofixed, dividing *C. tenuissimus* cells that clearly show valve SDVs.
2. Figure 6a only rudimentarily depict the previous model from Pickett-Heaps and needs to be amended. It needs to show the "fibrous band" that was supposed to guide and provide the force for expansion of the SDV. Interestingly this band was supposed to be actin based whereas microtubules were not supposed to be involved in the process. Interestingly, the cryoTEM data in the present work do not provide any evidence for actin filaments in the seta cytoplasm, but clearly demonstrate the presence of a microtubule parallel to the long axis of each seta. To avoid misunderstanding it should also be shown that the seta biosilica becomes exocytosed and thus is an extracellular structure, just like in the new model
3. What do the green arrows in Figures 5d and h point to?

Reviewer #2 (Remarks to the Author):

The revised manuscript by Gal et al has addressed my main recommendations relating to the transport pathway for Si in seta formation. The authors have carried out experiments with both varying levels of Si and the Si transport inhibitor Ge to show that seta formation is susceptible to both reduced Si levels and Ge. The authors have accordingly modified their hypothesis to one that proposes that the source of Si for extracellular Si deposition is intracellular rather than directly extracellular. There are, however some remaining issues that should be addressed. I consider that the manuscript has been improved and is better focused on possible mechanism.

1. In Fig 5, it is shown that reduced Si and Ge lead to inhibition of seta formation. The authors argue that there is differential sensitivity between seta and valve formation in support of their arguments that a SDV is not involved in seta Si deposition. However, I think that they should be cautious with this interpretation since the results presented in Fig 5 are non-quantitative and

indeed Si incorporation into the valve does appear to be reduced in both treatments. I think the most that can be concluded from these experiments is that formation of both setae and other silicified cellular components are disrupted by either reduced Si or the presence of Ge, indicating the involvement of an intracellular Si uptake pathway for both. Moreover, while I am satisfied that the authors provide good evidence that an SDV is not involved in seta Si deposition the results with reduced Si levels and Ge per se are not, as they stand, inconsistent with the role of an SDV in seta formation.

2. Following on from this, there is also a potentially confusing sentence in the Abstract (lines 20-22). The authors state that "In addition, the formation of these silica extensions is more susceptible to perturbations in the environmental conditions when compared to other cell wall elements, as expected for an extracellular silicification mechanism." The environmental perturbations relate to the availability of Si and are interpreted in terms of an intracellular transport pathway. As far as I can see there is no evidence (or discussion thereof) to suggest that the external precipitation per se is affected by environmental conditions.

3. I don't understand the labelling below Fig 5h. This needs to be better explained in the legend.

4. If vesicles were involved in delivering Si to the seta and the extracellular space, it is not their visible numbers in a snapshot that are important, rather it is their rate of turnover. It is not impossible to have high turnover rate with low steady state numbers. Furthermore, the argument presented in line 308-309 that "...as, in general, vesicular transport of silica building blocks is not established in diatoms" could also be applied to extracellular precipitation more generally. Perhaps a note to say that "while vesicular transport cannot be ruled out...." would provide a more balanced argument.

Reviewer #3 (Remarks to the Author):

I would like to thank the authors for their efforts in clarification, both in the response to the reviewers and in the text and for the new results they present, which shed additional light. This new version is more convincing even if the results are still sometimes over-interpreted. I will raise 3 points on this subject:

1) Absence of SDV

The fact that no SDV is seen in any of the samples does not mean that there is no SDV. Until now, the "absence of evidence is not proof of absence". As the response to the reviewers points out, this is "a working hypothesis that can inspire future work". The idea of a different silicification process for setae than for frustule is interesting, innovative and deserves to be examined. For this reason, I am in favour of the publication of this article, but I would like the authors to be much more moderate in what they say and to "suggest" rather than to "assert".

2) Source of silicic acid

I particularly appreciated the authors' modification of the hypothesis that silica was transported directly from the surrounding environment to the seta light by diffusion, which was highly implausible given the low concentrations in seawater compared to intracellular concentrations. In the new version, the author finally explains that silica would be transported via an intracellular route and concentrated by an active mechanism before being transported to the tip of the setae.... Finally, the uptake of silica would thus be intracellular at first (via SITs or by diffusion, therefore?), just as for the formation of the frustule cell wall. This implies that the regulation would be rather internal. I think this point could be clarified.

3) Influence of environmental conditions

(L21 in the abstract and later in the text) I am not convinced by the fact that setae formation is more influenced by environmental conditions than other cell wall elements.

The experiment does not allow a quantitative comparison of the impact of silica starvation on cell wall elements and on the setae. The authors argue using HAADF-STEM as a proxy for the Si content of the cell wall (L200). This is interesting, but this is the first time I hear about this

method as a quantitative tool. Has the method been validated by biogenic silica measurements (any refs)?

The fact that setae do not grow at low silica concentrations does not mean that the process of valve formation is different. It is well known that at low concentrations of silicic acid or in the presence of high concentrations of Ge, the structure of the frustule itself is modified, and it is not surprising to see anomalies occurring. The absence of setae could be one of them. If their formation occurs at the final stage, it is not surprising that they are more strongly impacted by the lack of silica. Nevertheless, this does not mean that the degree of regulation of extracellular processes is weaker than that of SDV (L260-265).

Response to Reviewers

Below is our detailed response (in blue) to the reviewers' reports (in black). In addition, a revised manuscript file, where all the changes are marked in red, was uploaded to the submission system.

Reviewer #1:

The revised title of the manuscript better reflects the current state of knowledge, because also the new data still do not provide clear-cut evidence for extracellular silica formation. The added experiment on seta formation in the presence of very low silicic acid concentration did not show "instantaneous" (i.e. during the next cell division) cessation of seta formation. Instead it took several generations before a reduction in the length and number of setae took effect. The authors therefore suggest that seta formation is fueled from intracellular silicic acid pools where silicic acid is concentrated, and then exported into the extracellular matrix where they believe the biomineralization of setae to take place. While this scenario seems feasible, it relies on using the general intracellular Si pool for seta formation. As a consequence, the experiment is unable to decouple Si use for biomineralization in the SDV from the use of Si for the proposed extracellular seta biomineralization. Therefore, the Si-limitation experiment can neither support nor disprove an extracellular biomineralization mechanism for seta formation.

We agree with this view. Indeed, providing a 100% proof for either intracellular or extracellular silicification of the setae will await future works. Nevertheless, we still hold that the fact that seta formation is decoupled from valve formation, even though the Si supply pathway is similar, at least gives additional credibility to the suggestion that they form by different mechanisms.

Nevertheless, the excellent cryoTEM data are sufficient enough to seriously question the previous model for seta formation (as was put forward by Pickett-Heaps), and the authors provide a feasible alternative model for this process. By changing the title and also throughout the text the authors no longer claim to have PROVEN the existence of extracellular silica biomineralization in diatoms. Instead they claim, justifiably so, that their data is consistent with extracellular biomineralization. I regard this paper as very important, and recommend it to be published after the following additions and modifications have been made:

We appreciate the positive feedback and are thankful for the constructive comments.

1. In the rebuttal letter the authors present an image from HP-frozen/freeze substituted diatom cell that contains silica filled SDVs. Apparently, the cell does not seem to be from the same diatom species that was used for the present study on seta development. The fact that a different

fixation method and a different species was used is not really a proof for the ability to visualize SDVs in cryofixed cells of *C. tenuissimus*. Therefore, the authors should provide in the supplementary information images from cryofixed, dividing *C. tenuissimus* cells that clearly show valve SDVs.

Studying SDVs using cryo electron tomography is an exciting direction that we are actively pursuing in the lab, not only for *C. tenuissimus* but also for other diatom species. However, this is an extremely challenging task, beyond the scope of the current work, since unlike the work on the setae, the cells need to be thinned for visualizing their center, where the SDV is located. We are currently in the process of establishing such an experimental pipeline. Namely, to synchronize cells, cryo-fix them at various time points, mill the cells with a cryoFIB to expose the SDV for tomography, and tomographic data collection. As mentioned before, this involves extremely challenging tasks at several stages of the process and therefore outside the scope of the current manuscript.

We would like to emphasize that the relevant point here is the ability of electron tomography to detect intracellular membrane-bound organelles (such as the SDV). cryoET is now a flourishing field in structural biology and detecting membranes is well within the abilities of this methodology. Current state-of-the-art of cryoET is nearing the ability to elucidate structures at a close to atomic resolution. In the data we present there are internal controls for the resolution of the methodology, e.g. the cell membrane that is present in the data sets, enabling the reader to conclude that lipid bilayers can be detected in our results. The image of the HPF and freeze-substituted *Thalassiosira pseudonana* cell that we included in the previous response letter demonstrated that even with methods with lower preservation power the SDV is clearly visible, when present, in sectioned cells. Thus, we believe that it is not the technical ability of cryoET to detect SDV, which is under question.

In order to emphasize that cryoET has the spatial resolution needed to resolve lipid bilayers membranes of organelles we revised the following sentences in the introduction and discussion:

Lines 54-55: "The high-resolution native-state information acquired using cryoET clearly resolve cellular structures such as lipid bilayer membranes."

Lines 266-268: "It is important to stress that the resolution of cryoET in general, and our data sets specifically, is sufficient to detect lipid bilayer membranes, for example, the cell membrane."

2. Figure 6a only rudimentarily depict the previous model from Pickett-Heaps and needs to be amended. It needs to show the "fibrous band" that was supposed to guide and provide the force for expansion of the SDV. Interestingly this band was supposed to be actin based whereas microtubules were not supposed to be involved in the process. Interestingly, the cryoTEM data in the present work do not provide any evidence for actin filaments in the seta cytoplasm, but

clearly demonstrate the presence of a microtubule parallel to the long axis of each seta. To avoid misunderstanding it should also be shown that the seta biosilica becomes exocytosed and thus is an extracellular structure, just like in the new model

Figure 6 was revised thoroughly. Panel a now shows all components of the previous model, including the 'fibrous band' and the need for an exocytosis event. Also panels b-f were revised to highlight the identified components in this study.

Figure 6 legend now reads: "a) An SDV-mediated process that includes SDV membrane surrounding the forming silica, and a dedicated cellular complex that is responsible for elongation of the seta (based on the works of J. Pickett-Heaps). Only after maturation the silica is exocytosed and becomes exterior to the cell membrane."

3. What do the green arrows in Figures 5d and h point to?

The green arrows indicate PDMPO labeled valves identified on the grid with a fluorescence microscope. By finding the same PDMPO-labeled valves in the TEM we could verify that they formed in the presence of Ge. This was poorly described in the previous version, and is now clearly explained in the main text and the figure caption:

Lines 229-232: "In order to identify the specific valves that were produced in the presence of Ge, PDMPO was added to the culture concurrently with Ge addition. PDMPO labeled valves were located on the TEM grid using a correlative optical-electron microscopy approach (Fig. 5d,h insets)."

Figure caption, lines 214-218: "Insets in (d,h) show optical microscopy images of the same cells as in the main panels. The fluorescence of PDMPO, which was added with Ge to the growth medium, allowed the identification of specific valves (indicated with green arrows) that formed in the presence of Ge. These valves do not have setae."

Reviewer #2 (Remarks to the Author):

The revised manuscript by Gal et al has addressed my main recommendations relating to the transport pathway for Si in seta formation. The authors have carried out experiments with both varying levels of Si and the Si transport inhibitor Ge to show that seta formation is susceptible to both reduced Si levels and Ge. The authors have accordingly modified their hypothesis to one that proposes that the source of Si for extracellular Si deposition is intracellular rather than directly extracellular. There are, however, some remaining issues that should be addressed. I consider that the manuscript has been improved and is better focused on possible mechanisms.

We would like to thank the reviewer again for both previous and current suggestions that considerably improved the paper.

1. In Fig 5, it is shown that reduced Si and Ge lead to inhibition of seta formation. The authors argue that there is differential sensitivity between seta and valve formation in support of their arguments that a SDV is not involved in seta Si deposition. However, I think that they should be cautious with this interpretation since the results presented in Fig 5 are non-quantitative and indeed Si incorporation into the valve does appear to be reduced in both treatments. I think the most that can be concluded from these experiments is that formation of both setae and other silicified cellular components are disrupted by either reduced Si or the presence of Ge, indicating the involvement of an intracellular Si uptake pathway for both. Moreover, while I am satisfied that the authors provide good evidence that an SDV is not involved in seta Si deposition the results with reduced Si levels and Ge per se are not, as they stand, inconsistent with the role of an SDV in seta formation.

We agree with the criticism raised by all of the reviewers that the data of disrupted seta formation in low Si and in the presence of Ge cannot by itself exclude the possibility of an SDV-mediated process. Nevertheless, and in agreement with the reviewers, we think that together with the cryoET data this decoupling between seta formation and valve formation supports the possibility of a different mechanism for seta formation.

The reviewer is right that the previous HAADF-STEM data in figure 5 was only semi-quantitative and the degree of low-Si effect on the valve silica was hard to judge. Since HAADF-STEM can be used as a quantitative tool with the right samples and imaging conditions, we carried out further experiments in order to provide quantitative comparisons between the different silica elements.

This further analysis does not change any fundamental issue regarding the underlying process, but we think it is better to judge the possible scenarios based on quantitative data. In the added experiments we grew again the cells in different Si concentrations, after an acclimation period we extracted and cleaned the valves from their organic coverage and cell debris (this step was not done in the previous experiments). We performed quantitative HAADF-STEM experiments that enabled us to compare the amount of Si in each treatment. The Si content at the 2.5 μ M sample was lower than at 5 μ M or 10 μ M, but similar to the sample grown at 330 μ M. This shows that also valve silicification is affected by the changing Si concentrations in the medium, but not in a correlated way to the complete cessation of seta formation.

The new data is referred to in the new Supplementary Figure 2, and in lines 205-209: "In order to relate the HAADF-STEM contrast to Si content in the valve, the cell walls were extracted by a mild oxidizing treatment that removes adherent organic material. A quantitative HAADF-STEM

imaging of such cell walls indeed shows a minor reduction of the silica content of the valves as Si concentrations of the medium were lowered (Supplementary Fig. 2).“

2. Following on from this, there is also a potentially confusing sentence in the Abstract (lines 20-22). The authors state that “In addition, the formation of these silica extensions is more susceptible to perturbations in the environmental conditions when compared to other cell wall elements, as expected for an extracellular silicification mechanism.” The environmental perturbations relate to the availability of Si and are interpreted in terms of an intracellular transport pathway. As far as I can see there is no evidence (or discussion thereof) to suggest that the external precipitation per se is affected by environmental conditions.

The sentence in the abstract was revised to clearly state the observed effect of low Si:
Lines 20-22: “In addition, the formation of these silica extensions is halted at low silicon concentrations that still support the formation of other cell wall elements, further alluding to a different silicification mechanism.”

3. I don't understand the labelling below Fig 5h. This needs to be better explained in the legend.

We revised Figure 5 caption to better explain the statistical data provided. The updated section in the caption now reads:

Lines 218-224: “For each experimental condition, a statistical summary of the average number and standard deviation of setae is presented below the images. In the different Si concentrations (a-c, e-g), the number of setae declines from the a value that is close to the ideal four setae per cell to almost no setae at all. In the Ge-treated culture (d,h), the valves that are labeled with PDMPO, thus the ones that formed in the presence of Ge, have almost no setae, while the unlabeled valves that form prior to Ge addition have the expected two setae per valve. Scale bars are 5 μm .”

4. If vesicles were involved in delivering Si to the seta and the extracellular space, it is not their visible numbers in a snapshot that are important, rather it is their rate of turnover. It is not impossible to have high turnover rate with low steady state numbers. Furthermore, the argument presented in line 308-309 that “...as, in general, vesicular transport of silica building blocks is not established in diatoms ...” could also be applied to extracellular precipitation more generally. Perhaps a note to say that “while vesicular transport cannot be ruled out...” would provide a more balanced argument.

We agree that such frozen snapshots cannot represent a process that might have a high turnover. The relevant text was revised to be more in-line with the current level of understanding and new findings:

Lines 326-333: "One possibility for silicon transport is a diffusion-based mechanism, primarily from the cell body along the seta. Such an intracellular diffusion process can rely on high intracellular Si concentrations that were observed in other species^{6,35,36}, and can supply the needed Si at the seta tip. An alternative mechanism is transport by Si-containing vesicles. However, only occasionally vesicles were observed inside forming setae, and their interior lacked distinct electron-dense material that could be a dense Si-phase. Therefore, while vesicular transport cannot be ruled out, we currently consider a diffusion-based mechanism more likely."

Reviewer #3 (Remarks to the Author):

I would like to thank the authors for their efforts in clarification, both in the response to the reviewers and in the text and for the new results they present, which shed additional light. This new version is more convincing even if the results are still sometimes over-interpreted. I will raise 3 points on this subject:

1) Absence of SDV: The fact that no SDV is seen in any of the samples does not mean that there is no SDV. Until now, the "absence of evidence is not proof of absence". As the response to the reviewers points out, this is "a working hypothesis that can inspire future work". The idea of a different silicification process for setae than for frustule is interesting, innovative and deserves to be examined. For this reason, I am in favour of the publication of this article, but I would like the authors to be much more moderate in what they say and to "suggest" rather than to "assert".

We fully agree that it is better to clearly differentiate observations from interpretations. Indeed, the question of how to prove an absence of something is at the heart of this paper, and since this is a delicate argumentation it warrants to be as impartial as possible.

We summarize here all revisions made to emphasize this point:

Title: already in the previous version we added "structural evidence" to be specific.

Abstract: the wording was revised in lines 19-22: "Remarkably, our data suggest that, contradictory to the ruling paradigm, these intricate structures form outside the cytoplasm. In addition, the formation of these silica extensions is halted at low silicon concentrations... further alluding to a different silicification mechanism."

Introduction: the wording was revised in line 56-57: "...outside the cell membrane, suggesting a formation process that is not mediated by an SDV."

Results: lines 183-184: "... provides a structural evidence that points to a silicification process that does not involve an SDV. Since the fact that we did not observe an SDV is not rationally

sufficient to prove its absence, we sought more evidence regarding the process of seta formation.”

Discussion: a detailed explanation in lines 266-273: “It is important to stress that the resolution of cryoET in general, and our data sets specifically, is sufficient to detect lipid bilayer membranes, for example, the cell membrane. Therefore, the fact that we do not observe any membrane-bound organelle that contains dense silica structures within actively silicifying setae strongly suggests that such silica deposition vesicles are not involved in the process. Based on the absence of any intracellular vesicle in which the seta forms, we deduce that this is an extracellular process, even though we are aware that a complete proof will need a comprehensive description of the process in action.”

A concluding sentence in lines 345-346: “...this work proposes an alternative mechanism,...”

2) Source of silicic acid: I particularly appreciated the authors' modification of the hypothesis that silica was transported directly from the surrounding environment to the seta light by diffusion, which was highly implausible given the low concentrations in seawater compared to intracellular concentrations. In the new version, the author finally explains that silica would be transported via an intracellular route and concentrated by an active mechanism before being transported to the tip of the setae.... Finally, the uptake of silica would thus be intracellular at first (via SITs or by diffusion, therefore?), just as for the formation of the frustule cell wall. This implies that the regulation would be rather internal. I think this point could be clarified.

We agree that even though the silicification process happens outside the cell membrane it is highly controlled and regulated by the cell. We added this point to the paragraph in the discussion that deals with Si transport. Since also the rest of the paragraph was revised along the comments of the other reviewers here is the update wording of the entire paragraph:

Lines 325-338: “An interesting open question regarding this process is the source of the silicon building blocks used to form the seta^{5,6,33,34}. One possibility for silicon transport is a diffusion-based mechanism, primarily from the cell body along the seta. Such an intracellular diffusion process can rely on high intracellular Si concentrations that were observed in other species^{6,35,36}, and can supply the needed Si at the seta tip. An alternative mechanism is transport by Si-containing vesicles. However, only occasionally vesicles were observed inside forming setae, and their interior lacked distinct electron-dense material that could be a dense Si-phase. Therefore, while vesicular transport cannot be ruled out, we currently consider a diffusion-based mechanism more likely. Regardless of the exact mechanism, our results suggest an intracellular Si-transport system that is under the control of the cell. Therefore even though seta formation occurs outside the cell membrane the process is still under a strict cellular control, from the confinement of the silicification volume, to the possible involvement of macromolecules, and the supply of the needed Si building blocks.”

3) Influence of environmental conditions (L21 in the abstract and later in the text) I am not convinced by the fact that setae formation is more influenced by environmental conditions than other cell wall elements.

The experiment does not allow a quantitative comparison of the impact of silica starvation on cell wall elements and on the setae. The authors argue using HAADF-STEM as a proxy for the Si content of the cell wall (L200). This is interesting, but this is the first time I hear about this method as a quantitative tool. Has the method been validated by biogenic silica measurements (any refs)?

Regarding the sentence in the abstract, also in accordance with point #2 of Reviewer #2, we revised it to specifically address the low Si experiment:

Lines 20-22: "In addition, the formation of these silica extensions is halted at low silicon concentrations that still support the formation of other cell wall elements, further alluding to a different silicification mechanism."

HAADF-STEM is a highly quantitative method. The inelastic scattering of the electrons is proportional to the Z-value of the elements in the beam path. Of course, in order to make it a quantitative comparison between different samples there is a need to keep identical acquisition parameters of the electron optics and the detector (which is detailed in the Methods). We added two references to line 203. One of them is demonstrating the quantitative abilities of HAADF-STEM, and the other is demonstrating how the intensity of HAADF-STEM imaging of mesoporous silica is in-line with simulations of silica densities (we are not aware of similar works with biogenic silica but for this issue the source of the material is irrelevant). In addition, we added a new data set (Supplementary Fig. 2), that shows quantitatively the small differences in Si content in the valves grown at the various Si concentrations of the medium. Please see the details in the response to comment #1 of Reviewer #2.

The fact that setae do not grow at low silica concentrations does not mean that the process of valve formation is different. It is well known that at low concentrations of silicic acid or in the presence of high concentrations of Ge, the structure of the frustule itself is modified, and it is not surprising to see anomalies occurring. The absence of setae could be one of them. If their formation occurs at the final stage, it is not surprising that they are more strongly impacted by the lack of silica. Nevertheless, this does not mean that the degree of regulation of extracellular processes is weaker than that of SDV (L260-265).

The choice of words in the last sentence was poor. While the regulation of intracellular and extracellular processes is different in some aspects, there is no place for ranking their degrees. We modified the last sentence to be in line with the entire notion:

Lines 273-282: “Being a process that occurs in the extracellular environment does not mean it is happening spontaneously in bulk seawater without cellular regulation. The organic coat that surrounds the seta creates a confined nanoscale environment, which can be chemically very similar to the conditions inside an SDV. It is plausible that the cell membrane transfers the needed organic and inorganic building blocks into this extracellular space similarly to the transport mechanisms into an SDV. Finally, the observations that under silicification-limiting conditions seta development stops before SDV-mediated processes suggest that the regulation of the extracellular process is different, at least in some aspects, from that of the SDV.”

REVIEWERS' COMMENTS

Reviewer #1 (Remarks to the Author):

The revised version of the manuscript has resolved my concerns as well as - in my view - those of the two other reviewers.

I recommend to publish the manuscript in its current form.

Reviewer #2 (Remarks to the Author):

The authors have largely addressed my remaining concerns in the revised manuscript. Essentially the conclusions are that there is no structural evidence for the involvement of an SDV in seta but an intracellular route for delivery of Si is very likely. However, there appears to be differential sensitivity of seta or valve formation to the availability of Si. As acknowledged, while each observation alone is not sufficient evidence to conclude a lack of SDV involvement in seta formation, together they do point in that direction. Further exploration of the underlying mechanism of seta formation would of course be very useful in providing more definitive evidence. For example, the authors could consider some simple experiments with inhibitors of vesicle transport (e.g. brefeldin A) that could potentially distinguish between exocytotic and diffusion mechanisms for supplying Si to the forming seta.